# *Gluconacetobacter diazotrophicus* Pal5 Enhances Plant Robustness Status under the Combination of Moderate Drought and Low Nitrogen Stress in *Zea mays* L.

**DOI:** 10.3390/microorganisms9040870

**Published:** 2021-04-17

**Authors:** Muhammad Aammar Tufail, María Touceda-González, Ilaria Pertot, Ralf-Udo Ehlers

**Affiliations:** 1Department of Civil, Environmental and Mechanical Engineering, University of Trento, Via Mesiano 77, 38123 Trento, Italy; 2e-nema Gesellschaft für Biotechnologie und Biologischen Pflanzenschutz mbH, Klausdorfer Str. 28-36, 24223 Schwentinental, Germany; m.touceda@e-nema.de (M.T.-G.); ehlers@e-nema.de (R.-U.E.); 3Department of Sustainable Agro-Ecosystems and Bioresources, Research and Innovation Centre, Fondazione Edmund Mach, Via E. Mach 1, 38098 San Michele all’Adige, Italy; ilaria.pertot@unitn.it; 4Center Agriculture Food Environment (C3A), University of Trento, Via E. Mach 1, 38098 San Michele all’Adige, Italy

**Keywords:** combined abiotic stress, plant growth promoting bacteria (PGPB), endophytes, diazotrophs, *nifH* gene, N fixation

## Abstract

Plant growth promoting endophytic bacteria, which can fix nitrogen, plays a vital role in plant growth promotion. Previous authors have evaluated the effect of *Gluconacetobacter diazotrophicus* Pal5 inoculation on plants subjected to different sources of abiotic stress on an individual basis. The present study aimed to appraise the effect of *G. diazotrophicus* inoculation on the amelioration of the individual and combined effects of drought and nitrogen stress in maize plants (*Zea mays* L.). A pot experiment was conducted whereby treatments consisted of maize plants cultivated under drought stress, in soil with a low nitrogen concentration and these two stress sources combined, with and without *G. diazotrophicus* seed inoculation. The inoculated plants showed increased plant biomass, chlorophyll content, plant nitrogen uptake, and water use efficiency. A general increase in copy numbers of *G. diazotrophicus*, based on 16S rRNA gene quantification, was detected under combined moderate stress, in addition to an increase in the abundance of genes involved in N fixation (*nifH*). Endophytic colonization of bacteria was negatively affected by severe stress treatments. Overall, *G. diazotrophicus* Pal5 can be considered as an effective tool to increase maize crop production under drought conditions with low application of nitrogen fertilizer.

## 1. Introduction

The main limiting environmental factors influencing maize production worldwide are drought and low N stress [1,2,3]. Drought stress cause up to a 15% annual yield loss in maize, which is gradually increasing due to climate change [4,5]. In the current situation, agriculture is significantly impacted by climate change, causing a global threat to food security [6]. This is mostly derived from the loss of arable land due to drought stress, land degradation, and environmental restrictions on agricultural production as reported by the UN General Assembly [7]. In the 20th century, chemical synthesis of nitrogen fertilizer through the Haber–Bosch process improved agriculture production and food security [8]. Owing to the instability of synthetic nitrogenous fertilizers, over half of the world’s nitrogen fertilizer is lost to leaching in groundwater and volatilization in the atmosphere in the form of nitrous oxide, a potent greenhouse gas [9,10]. Another leading agricultural challenge is to supply adequate and sufficient nitrogen to cereal crop plants. Cereals contain 75% carbohydrates and up to 15% protein, contributing 50% in global terms of energy supply [11]. 

Plants face diverse biotic and abiotic stresses in hostile environments. Drought stress has been counted as a critical issue that negatively affects plant growth in different developmental stages, and, more importantly, crop yield [12]. Several approaches are employed to enhance drought tolerance and nitrogen use efficiency in plants with higher yields. Current agricultural production approaches are costly and non-renewable, e.g., improper use of chemical fertilizers can contribute to greenhouse gas emissions and cause various environmental problems [13,14].

Plants have natural mechanisms for defending against multiple stresses and one of them is the synergic interaction with microorganisms [15]. Such plant-beneficial microorganisms, specifically bacteria, offer several advantages to their host plants and allow them to withstand various biotic and abiotic stresses that can have detrimental effects on their growth and development [16,17,18]. Endophytic bacteria that, directly or indirectly, support plant growth, development, and health status are usually known as endophytic plant growth promoting bacteria (PGPEB). PGPEBs can increase productivity and confer plant immunity and systemic resistance to abiotic stresses that can induce physiological, molecular, and biochemical changes in plants. These PGPEBs improve osmotic adjustment, phytohormone regulation, nutrient (N, P, K etc.) acquisition, enzymatic and non-enzymatic antioxidants’ activation mechanisms, and osmo-protectants’ production [19,20,21]. Particularly, endophytic diazotrophic bacteria are of special interest since they are capable of fixing atmospheric nitrogen, entrapping N_2_ and converting it in NH_3_, a form that is readily utilized by plants [22]. This process is catalyzed by the oxygen-sensitive enzyme nitrogenase, formed by various subunits encoded by the *nifH*, *nifD*, and *nifK* genes. Among these three genes, *nifH* has become the most used reference marker in studies of diversity and abundance of nitrogen-fixing microorganisms [23]. Furthermore, association with nitrogen-fixing PGPEBs may increase the leaf nitrogen concentration of plants which is essential to synthesize chlorophylls, nucleic acids, and proteins [24]. Some nitrogen-fixing endophytes are being currently tested as biofertilizers, and these bacteria include members of the genera Azoarcus, Achromobacter, Burkholderia, Gluconoacetobacter, Herbaspirillum, Klebsiella and Serratia [25]. *Gluconacetobacter diazotrophicus* Pal5 is an endophytic diazotrophic bacteria, which has previously been reported in ameliorating the effects of drought stress in rice and sugarcane plants and promoting plant growth in low nitrogen environments [26,27,28]. 

Maize (*Zea mays* L.) is a multipurpose crop with wide adaptability to different agro-climatic conditions. It is grown in most parts of the world and is preferred by farmers because it is a C4 cash crop with a high photosynthetic rate and grain production potential, with a dual-purpose use as food source (grain and fodder) and raw material for industry [29,30,31,32].

Therefore, the aim of the present study was to investigate the potential of the diazotrophic bacteria endophyte, *Gluconacetobacter diazotrophicus* Pal5, for improving the growth of maize plant under individual and combined effects of drought and low nitrogen stress. Our findings suggests that use of endophytic diazotrophic bacteria can be a promising solution to improve plant tolerance to a combination of multiple stresses.

## 2. Materials and Methods

### 2.1. Experimental Design

#### 2.1.1. Inoculum Preparation

The bacterial strain *Gluconacetobacter diazotrophicus* Pal5 (Gd) type strain (DSM 5601) was ordered from DSMZ-German Collection of Microorganisms and Cell Cultures GmbH, Leibniz Institute, Germany (https://www.dsmz.de/, accessed on 6 March 2019). The bacterium was cultured in 250 mL Sabouraud 2% Glucose (SG) broth at 28 °C for 48 h at 180 rpm in an orbital shaker. The culture optical density was measured at λ = 600 nm using a spectrophotometer and adjusted to 0.1 to obtain a uniform population of bacteria, 10^8^ colony forming units (CFU) mL^−1^, for inoculation.

#### 2.1.2. Pot Experiment Setup

A pot experiment was conducted in the growth room to evaluate the effectiveness of the nitrogen fixing bacterial strain Gd for promoting growth and yield of maize under nitrogen and water deficit conditions. Maize seeds (variety: KXB 5146, batch number: 16V-4053, KWS, Einbeck, Germany) were surface sterilized before inoculation, according to Naveed et al. [33]. Sterile maize seeds were incubated in a 10^8^ CFU mL^−1^ of SG broth overnight culture of Gd for 2 h. Untreated seeds were maintained for 2 h in sterile 2% SG broth. Five seeds, either inoculated or not, were sown in plastic pots containing 550 g of sterile commercial soil with fewer amount of available nutrients (50–200 mg L^−1^ nitrogen, 80–150 mg L^−1^ P_2_O_5_ and 150–300 mg L^−1^ K_2_O; Gardol). Two days after germination, the plants were trimmed to one. Plastic pots were sterilized using 70% ethanol and soil was sterilized two times at 121 °C for 40 min. Prior to seed sowing, an equal amount of sterile distilled water was applied to the pots to maintain optimal soil moisture. Temperature was set to 25 ± 2 °C, the photoperiod to a 16 h light and 8 h dark with 36% humidity. 

In total, 6 treatments were set up (Table 1). The experiment comprised three levels of nitrogen and three levels of drought stress. The soil moisture regimes were: 35% (Dr.35), 50% (Dr.50), and 100% (Dr.100) of soil water holding capacity (WHC), representing severe, moderate, and no water stress conditions, respectively. Soil water content regimes were controlled by weighing the pots and irrigating the plants during the experimental period starting from 12 days after sowing. The nitrogen treatments consist of no-N (N-Free, 0 mg N pot^−1^), 50% N (N-50, 150 mg N pot^−1^), and 100% N (N-100, 300 mg N pot-1) of recommended Nitrogen dose. Nitrogen doses were applied in the form of modified Hoagland solution, 7 days after sowing. A modified Hoagland solution was prepared using calcium nitrate, as a source of nitrogen fertilizer [34]. An N-100 dose was calculated based on nitrogen, phosphorous, and potassium (N-P-K: 160-100-60 kg ha^−1^) fertilization recommended by Naveed et al. [33]. Five replicates per treatment were set up, using either untreated or Gd inoculated seeds, making a total of 60 experimental units.

### 2.2. Plant Analysis

Maize plants were harvested 26 days after sowing and taken from all five pots in each treatment. Bulk soil attached to the roots was removed by gently shaking and followed by a water rinse. Roots, stems, and leaves of each plant were separated for further analysis. Samples needed for molecular analysis were immediately stored at −80 °C. 

The following parameters related with the plant were measured: plant biomass, chlorophyll content and relative water content, plant water consumption and efficiency, and leaf rolling score. Shoot and root weight were measured with a weighing balance. Plant images were captured with a centimetre scale and analysed with the open access software platform FIJI (ImageJ) [35]. Plant growth parameters such us shoot length, root length, stem diameter, and leaf width were measured. Leaf relative water contents (RWC) were calculated according to [36]. To evaluate the photosynthetic efficiency, *Chl a*, *b* and carotenoids were measured. Therefore, 0.5 g of fresh leaf cut from the middle part of the older leaves was ground in 4.5 mL acetone (80%) using a porcelain mortar and then centrifuged at 3000 rpm for 5 min. The mixture was brought to the volume of 20 mL by adding distilled water. The final solution was exposed to a wavelength of 646 and 663 nm to determine the concentration of *Chl a* and *b*, respectively, and 470 nm for carotenoids using a spectrophotometer. Chlorophyll concentration per mg of fresh weight was determined based on the method described by Lichtenthaler and Wellburn (1983). Plant shoot samples, three replicates per treatment, were sent to LUFA^®^ (https://www.lufa-nord-west.com/, accessed on 31 August 2020) for nitrogen analysis. Then, shoot nitrogen uptake and nitrogen use efficiency were calculated as described by [37]. Plant water consumption (PWC), i.e., the total evapotranspiration from maize plant and soil, was calculated from the water balance in each experimental pot according to Wang et al. [38]. For the whole plant, water use efficiency (WUE) was calculated as the ratio between shoot dry matter (DM) and PWC during the experimental period. The leaf rolling score included five levels: 1, leaf is unrolled and turgid; 2, leaf rim starting to roll; 3, leaf has a shape of a ‘V’; 4, rolled leaf rim covers part of leaf blade; and 5: leaf is rolled like an onion [3].

### 2.3. DNA Isolation

Root, stem, and leaf samples from three plants per treatment were used for DNA isolation. Namely, 0.5 g of tissue were cut and sterilized with 70% ethanol for 30 s, treated with 2% NaClO for 10 s, and followed by 3 times washing with sterile distilled water for 1 min each. Surface sterilized samples were grounded in liquid nitrogen using autoclaved pistil and mortar. Finally, DNA was isolated from the grounded plant samples using the PureLink™ Microbiome DNA Purification Kit (Thermo Fisher Scientific GmbH, Dreieich, Germany), according to the manufacturer’s instructions. Pure DNA was stored at −20 °C until further needed.

### 2.4. G. diazotrophicus Pal5 Detection

In order to detect *G. diazotrophicus* in roots, stems, and leaves of the inoculated maize plants, a nested PCR approach was implemented. In the first PCR round, a primer pair targeting the whole 16S rRNA gene was used, the 16S-27F (5′-AGAGTTTGATCMTGGCTCAG-3′) and16S-1492R (5′-TACGGYTACCTTGTTA CGACTT-3′) [39]. In the second PCR run of the nested approach, the bacterial primer pair PAl5F2 (5′-GGCTTAGCCCCTCAGTGTCG-3′) and PAl5R2 (5′-GAAACAGCCATCTCTGACTG-3′) was used to amplify 16S rRNA gene fragments of *G. diazotrophicus* [40]. For the first PCR round, the reaction mixture (50 mL) contained: 1 µL DNA template, 1.25 Units DreamTaq DNA polymerase (Thermo Fisher Scientific, Germany), 1× DreamTaq Buffer, 0.2 mM of each dNTP, 0.125 µM of each primer. In the second PCR round, for 50 mL reactions, the following reactive concentrations were used: 1 µL PCR product, 1.25 Units DreamTaq DNA polymerase (Thermo Fisher Scientific, Germany), 1× DreamTaq Buffer, 0.2 mM of each dNTP, 0.125 µM of each primer. The gene fragments were amplified with a Mastercycler^®^ gradient thermocycler (Eppendorf, Hamburg, Germany). Thermal cycling conditions in both PCR rounds were: initial denaturation step of 95 °C for 5 min, followed by 35 cycles of denaturation at 95 °C for 1 min, annealing at 58 °C for 1 min and extension at 72 °C for 1 min and a final extension step for 7 min at 72 °C. The presence and correct size of PCR product was checked in a 1.5% agarose gel. The 16s rRNA gene fragments derived from the second PCR run were purified and sent to sequenced, in Starseq^®^ (Mainz, Germany), to confirm that DNA from *G. diazotrophicus* was amplified.

### 2.5. Design of Novel nifH Primers and Validation

In order to detect the *nifH* gene in the inoculated plants tissue, the *nifH* universal primer pair designed by Ueda et al. [41] was used. After PCR run, no amplification was detected. Therefore, a new primer pair was designed to detect *nifH* gene in *G. diazotrophicus* Pal5 specifically. Moreover, *G. diazotrophicus* Pal5 genome was compared with the universal *nifH* primers proposed by Ueda et al. [41], using the QIAGEN CLC Genomics workbench (QIAGEN, Hilden, Germany). The Gd genome regions, where the *nifH* universal primers were attached in silico, were the sequences used to design the new primer pair for this study. The novel primer pair Gd-nifH-F (5′-GCCTTTTATGGAAAGGGAGG-3′) and Gd-nifH-R (5′-AAGCCGCCGCAGACCACGTC-3′) were used to amplify *nifH* gene in the inoculated plants’ root, stem, and leaf tissues. For 50 mL PCR reactions, the following concentrations were used: 1 µL DNA template, 1.25 Units DreamTaq DNA polymerase (Thermo Fisher Scientific, Germany), 1X DreamTaq Buffer, 0.2 mM of each dNTP, 0.45 µM of each primer. The gene fragments were amplified with a Mastercycler^®^ gradient thermocycler (Eppendorf, Hamburg, Germany). PCR conditions consisted of an initial denaturation at 94 °C for 5 min, which was followed by 40 cycles of 94 °C for 50 s, annealing at 62 °C for 45 s, extension at 72 °C for 1 min, and the final extension for 7 min at 72 °C. The presence and correct size of PCR product was checked in a 1.5% agarose gel and verified by sequencing.

### 2.6. Quantification of nifH and G. diazotrophicus Pal5 16S rRNA Genes in Plant Tissues

The abundance of the *nifH* and the *G. diazotrophicus* 16S rRNA genes was assessed by quantitative PCR (qPCR). The qPCR was carried out with 2× GoTaq^®^ qPCR Master Mix containing a low level of carboxy-X-rhodamine (CXR) reference dsDNA-binding dye (Promega, Walldorf, Germany) on an Applied Biosystems StepOne™ and StepOnePlus™ Real-Time PCR Systems (ThermoFisher Scientific, Bremen, Germany). The oligonucleotide primer pairs used were PAl5F2/PAl5R2 and Gd-*nifH*-F/Gd-*nifH*-R (see above) at a concentration of 333 nM. The thermal cycling conditions for *G. diazotrophicus* 16S rRNA genes were one DNA-denaturation step at 95 °C for 5 min, followed by 40 cycles of 95 °C for 1 min, 58 °C for 1 min, and 72 °C 1 min. For *nifH* genes, the qPCR conditions were one cycle at 94 °C for and then continued with 40 cycles of 94 °C for 50 s, 62 °C for 45 s, 72 °C for 1 min. The 10-log-fold standard curves were produced as follows: *G. diazotrophicus* Pal5 DNA was used as a template for conventional PCR amplification of the *nifH* and 16S rRNA genes (see above, Section 2.5 and Section 2.4, respectively). The PCR products, with the expected size, were purified with the QIAquick PCR Purification Kit (Qiagen, Hilden, Germany), quantified with NanoDrop 2000c (Thermo Scientific, Wilmington, DE, USA), and the gene copy numbers were calculated with scienceprimer (http://scienceprimer.com/copy-number-calculator-forrealtime-pcr, accessed on 6 August 2020). Ten-fold serial dilutions of *nifH* and *Gd 16S rRNA* genes PCR products were prepared and used to generate the qPCR standard curve. 

The quantification of these genes in plants’ roots, stems, and leaves was carried out with 1 mL of DNA template added to the PCR master mix in 96-well plates. Negative controls without DNA template and standards were included in all plates, and the melting curves were evaluated to confirm the purity of the amplified products.

### 2.7. Statistical Analysis

The statistical analysis was performed using the R (version 4.0.4) package agricolae (version 1.3-3). After corroborating the normality and homogeneity assumptions, a one-way ANOVA was performed followed by a Tukey’s HSD test (α = 0.05). Data of qPCR were analyzed using StepOne^TM^ software v. 2.3. A regression analysis was used to determine the relationships between the measured parameters. The graphs were designed using a ggplot2 package in the R environment and Microsoft Excel.

## 3. Results

### 3.1. Effect of G. diazotrophicus Inoculation on Maize Plant Growth

In this study, significant differences were observed in fresh and dry weights of root and shoot parts of the inoculated plants as compared to non-inoculated plants (Figure 1a–d). Under severe drought (T1) and N deficiency (T3), no significant differences were observed in shoot and root weights (fresh and dry) between untreated and inoculated plants. However, when water holding capacity was at 50% (T2), i.e., moderate drought stress, Gd inoculation significantly increased the shoot fresh weight by 65%, shoot dry weight by 67%, root fresh weight by 30%, and root dry weight by 80% of maize plants (Figure 1a–d). Shoot fresh weight of maize plants increased by 66% with Gd inoculation when grown under medium N deficiency (T4) (Figure 1a), but no differences were seen in root fresh weight and shoot and root dry weights (Figure 1b–d). On the other hand, a significant increase of 28% in root fresh weight was observed in Gd inoculated plants under severe combined stress (T5) (Figure 1c). The highest increase in plant shoot fresh weight was observed in the moderate combined drought and nitrogen stress (T6) that modulated from 9 ± 2 g to 24 ± 3 g (Figure 1a). The shoot dry weight was also increased from 560 ± 99 mg to 1490 ± 234 mg in T6 treatment) (Figure 1b). However, no significant differences were observed in root weights in moderate drought and nitrogen stress treatment combined (T6; Figure 1c,d).

Gd inoculated plants showed an increase in shoot length when grown under moderate drought stress (50% WHC, T2), severe nitrogen stress (T3), and moderate combined stress (T6) (Figure 2a). The greatest increase in shoot length occurred when the plants were subjected to moderate drought and nitrogen stress (T6) and raised from 254 ± 68 cm in untreated plant to 385 ± 43 cm in Gd-inoculated plants (Figure 2a). In the same treatment, T6, the largest root sizes were also found, with a length of 170 ± 14 cm in untreated plants and 416 ± 95 cm for inoculated plants, meaning a 145% increase (Figure 2b). In addition, Gd inoculation caused an increase of 46% in root length when maize plants were grown under moderate nitrogen stress (150 mg N pot^−1^, T4) (Figure 2b). Whereas no differences were observed in other treatments in shoot and root lengths of inoculated maize plants as compared to untreated controls (Figure 2a,b).

In plants inoculated with *G. diazotrophicus*, an increase was shown in the leaf relative water content (RWC) of maize plants by 8%, 10%, and 6% under moderate water stress (50% WHC; T2), moderate nitrogen stress (150 mg N pot^−1^; T4), and moderate combined stress (50% WHC and150 mg N pot^−1^, T6) treatments (Figure 3). In other treatments, no significant differences were observed in Gd inoculated plants as compared to untreated control (Figure 3).

### 3.2. Plant Photosynthetic Efficiency

Differences in chlorophyll (a and b) contents between inoculated and untreated plants were found when the maize plants were grown under moderate drought stress (T2), moderate nitrogen stress (T4), and moderate combined stress (T6) (Figure 4a–c). The largest increase in chlorophyll content occurred under individual moderate nitrogen stress (T4), going from 2.8 ± 0.11 to 3.5 ± 0.12 mg·g^−1^ fresh weight (FW) in chlorophyll a and from 1.4 ± 0.04 to 1.7 ± 0.06 mg·g^−1^ FW in chlorophyll b (Figure 4a,b). Carotenoid contents, under moderate nitrogen stress (T4) and moderate combined stress (T6), were increased by 22% and 28%, respectively, when maize plants were inoculated with Gd (Figure 4d).

### 3.3. Nitrogen Contents in Plants and NUE

*Gluconacetobacter diazotrophicus* inoculated maize plants showed a significant increase in nitrogen content in shoots, as compared to untreated control, when growing under moderate drought stress (T2), severe N deficiency (T3), severe combined stress (T5), and moderate combined stress (T6). The highest increase in shoot nitrogen uptake was observed in T6 treatment, combined stress with 150 mg N pot^−1^ and 50 % WHC. This increment was 1.73 times higher in the inoculated plants than in the untreated control (Figure 5). 

Nitrogen use efficiency (NUE) was significantly increased in plants inoculated with Gd when growing under moderate drought stress (T2), moderate N deficiency (T4), severe combined stress (T5), and moderate combined stress (T6), as compared to untreated control plants. The increments in NUE were 50%, 163%, 204.05%, and 274%, respectively. No significant differences were observed under severe drought and N stress treatments (T1 and T3) (Figure 6).

### 3.4. Plant Water Consumption, Water Use Efficiency, and Leaf Rolling Scores

In terms of plant water consumption (PWC), the highest values were obtained in maize plants growing under severe and moderate N stress (T3 and T4), in both untreated and Gd inoculated (Table 2). Plants were well irrigated in these two treatments. However, the PWC of maize plants significantly increased under severe and moderate drought stress (T1 and T2) and under moderate combined stress (T6) when inoculated with Gd as compared to untreated controls (Table 2). Additionally, the highest increase in PWC was observed in T6, by 3.6% (Table 2). Plant water use efficiency (WUE) generally increased when maize plants were inoculated with Gd as compared to the untreated ones (Table 2). The highest WUE levels were observed in the T6 treatment, when the plants were inoculated and growing under moderate combined stress (Table 2). Leaf rolling is one of the main plant reactions against drought stress in maize crops. There was a clear reduction in the leaf rolling scores in plants growing under moderate drought stress (T2) and severe and medium combined stress (T5 and T6) when Gd was inoculated as compared to the untreated controls (Table 2).

### 3.5. nifH and G. diazotrophicus 16S rRNA Genes Abundance in Plant Tissues

The presence of *G. diazotrophicus* in roots, stems, and roots was confirmed by nested PCR targeting the *16S rRNA* gene. Regarding the qPCR results, the assays were highly reproduceable, and the standard errors were very low in the case of leaf and root tissues; however, stem tissues showed high standard errors making all the samples significantly similar to each other. The Gd 16s rRNA gene abundance ranged from 3.57 ± 0.11 (copy n° in T1) to 5.35 ± 0.14 gene copy number (log10) g^−1^ of fresh plant tissue (copy n° in T4). *G. diazotrophicus* 16s rRNA gene copy numbers were higher in inoculated plants growing under moderate N fertilization (T4) ranging from 4.31 ± 0.22 to 5.35 ± 0.11 gene copy number (log10) g^−1^ and moderate combined stress going from 4.59 ± 0.35 to 5.17 ± 0.15 gene copy number (log10) g^−1^ of fresh plant tissue (T6) (Figure 7). Moreover, in leaf tissues, the abundance of this gene was significantly higher in treatments T2 (moderate drought stress), T4, and T6 (Figure 7). No significant differences were observed in stem tissues; however, T5 showed least *Gd 16S rRNA* gene copy numbers as compared to other treatments. On the other hand, in *G. diazotrophicus* inoculated samples from T6 treatment, moderate combined stress sources, significantly higher *nifH* gene copy numbers were found in root and leaf tissues, as compared to the other treatments (Figure 7). No significant differences were observed in *nifH* gene abundance among treatments in stem tissues (Figure 7).

## 4. Discussion

The main limiting environmental influences in maize production worldwide is drought and low nitrogen stress, which have received considerable attention in recent years [1,3]. Plants growing under drought and low nitrogen stress display a series of physiological, biochemical, and genetic changes which have an adverse effect on plant growth and production [3,42]. Evolutionary plants have evolved mechanisms to cope with environmental unfavorable conditions, a process widely known as stress resilience [43,44]. Many studies found that plant growth promoting endophytic bacteria (PGPEB) can improve plant resilience to water deficit conditions [21]. Moreover, nitrogen fixing PGPEB can give dual benefits to plants, improving drought tolerance and supplying fixed nitrogen, for their better growth [45]. Few studies are currently available in the literature that investigate the effect that PGPEB inoculation has on the plant under two sources of combined stress [46]. However, the role of diazotrophic endophytic bacteria to relieve the combined effects of both stresses, drought, and low nitrogen has not been studied. Results obtained in this study showed that *Gluconacetobacter diazotrophicus* Pal5 strain may better colonize maize plants under moderate drought, low nitrogen, and combined stress based on the *nifH* and Gd 16S rRNA genes analysis. This might be due to the tolerance mechanism of *G. diazotrophicus* decreasing the level of drought stress [47].

In general, the fresh and dry weight of the root and shoot is greater in *G. diazotrophicus* inoculated plants. Under severe drought and nitrogen stress, the differences are not significant, while, at intermediate stress levels, 50% water holding capacity and 50% nitrogen addition, either in combination or individually, the inoculated bacteria have an effect on plant weight. *G. diazotrophicus* Pal5 is known to produce auxin phytohormones, activate plant defense mechanisms against abiotic stresses, and fix atmospheric nitrogen while living inside the plants [45,48]. A *Gluconacetobacter diazotrophicus* strain was reported to produce indoleacetic acid, a molecule which is active in tricarboxylic acid cycle expression, glyoxylate shunt and amino acid biosynthesis, contributing to the induction of plant growth [49]. The inhibitory effects of moderate drought and low nitrogen stress on Gd inoculated plants may have been ameliorated via hormonal action and/or nitrogen availability to the plants as suggested by Egamberdieva et al. [50]. Nitrogen is an essential plant nutrient that affects plant growth and metabolic pathways including photosynthesis [45]. Increase in shoot and root growth can possibly be explained by the increment on N availability in shoot when the plants were inoculated with *G. diazotrophicus*. Shirinbayan et al. [51] reported that nitrogen fixing plant growth promoting bacteria *Azotobacter* strains increased the nitrogen concentrations and shoot dry weight, shoot length, chlorophyll content, and water use efficiency in maize plant under drought stress at 40% field capacity. Similarly, Gd inoculation increased the shoot N uptake by 1.3 to 2.1 times, nitrogen use efficiency by 1.1 times to 3.8 times, water use efficiency by 2.31 to 2.88 times, and plant water consumption by 1.03 to 1.05 times. This might be due to an increase in N-fixation efficiency and phytohormone production ability of the endophyte. The *nifH* gene in *G. diazotrophicus* is involved in the nitrogen fixing process [52,53]. Therefore, a higher number of *nifH* gene copy numbers in leaves and stems might be responsible for more N-fixation in moderately stressed treatments. Furthermore, it was shown that auxins promote cell elongation and formation of lateral roots and root hairs. Hence, the stimulation of maize plant growth might be due to the endogenous auxins and the Gd produced auxins inside the plants [54]—thus resulting in increased length and biomass to absorb more water and nutrients from the soil. This is in accordance with our results: Gd-treated plants show higher shoot and root length at moderate levels of drought stress, either individually or in combination with N deficiency. This is similar to the findings of Sandhya et al. in [55], where *G. diazotrophicus* inoculation improved shoot length of maize plants. Ref. [56] have shown that, when root biomass and length increased conjointly, the water uptake of plants increased, and, therefore, the hydration status of leaves was higher when the intensity of stress conditions was lower. A larger and denser root system will not only influence the nutrient uptake, as described above, but also the water uptake [57]. In our study, the leaf relative water content was higher when the inoculated plants were grown under moderate drought and N stress conditions, but there was no effect of inoculation under severe water stress and N-free treatments. Several studies have found that drought and low nitrogen stress can negatively affect root length, morphology, and biomass. Peng et al. [58] demonstrated that N deficiency suppressed the lateral root growth and increased root death causing a decrease in root length but increased N supply increased the toot length and biomass. In accordance with this, Gd inoculated plants in moderate N stress treatments (T4 and T6) showed increased root length as compared to untreated plants. In addition, increased nitrogen supply by bacteria might be the reason for increased root length. Our results further indicate the importance of interactive effects of drought and low nitrogen stress on root length, biomass, and morphology with and without Gd inoculation. In the cases of plants under severe drought stress, plants close their stomata to avoid water loss by transpiration and preserve more water inside plants, in order to sustain the water deficit condition [59]. Interestingly, growth of maize plants and N-uptake is much lower in the N-free treatment than in the 50% nitrogen dose treatment, as the plants clearly showed lower leaf rolling in the former. This was mainly due to the weaker growth and longer root systems [60]. The main function of abscisic acid is to control the stomatal closure under drought stress conditions preventing plant water loss [61]. Cytokinins are involved in cell division and improve photosynthesis of the plant [62]. In this sense, maize plants inoculated with *G. diazotrophicus* Pal5 may have downregulated the abscisic acid concentration in stomatal cells or upregulated the levels of cytokinin, thereby regulating the stomal closure and photosynthesis [63]. In accordance with Gururani et al., we found that the net photosynthesis activity was reduced due to drought stress, and inoculation with PGPEB increased the photosynthetic rate of plants. Similarly, our findings indicate that photosynthetic efficiency is lower in plants growing under severe drought and nitrogen stress conditions. Therefore, Gd presence in the shoot might be involved in increasing the chlorophyll contents in moderately stressed plants, leading to increased pumping of photosynthesis-generated glucose, which is needed for plant growth processes [64]. Chlorophyll and carotenoids are products of the photosynthesis that are directly involved in sugar synthesis in plants [65]. Sugars and carbohydrates play important roles in signaling and defending stressed plants as they are the primary building framework and energy supply for the processing and maintenance of biomass [66]. These observations of increasing chlorophyll and carotenoids contents are in accordance with the previous reports on the use of PGPEB for improving plant tolerance to drought stress [33,55].

## 5. Conclusions

In conclusion, *G. diazotrophicus* Pal5 was shown to ameliorate the individual and combined effects of drought and low nitrogen stress from maize plants, by regulating plant defense mechanisms. Furthermore, it has the potential to promote maize plant growth under water deficit conditions and with low nitrogen application, thus it could be used effectively in sustainable agriculture. Therefore, seed inoculation with *G. diazotrophicus* can be a very successful tool for inducing individual and combined stress tolerance in maize plants.

## Figures and Tables

**Figure 1 microorganisms-09-00870-f001:**
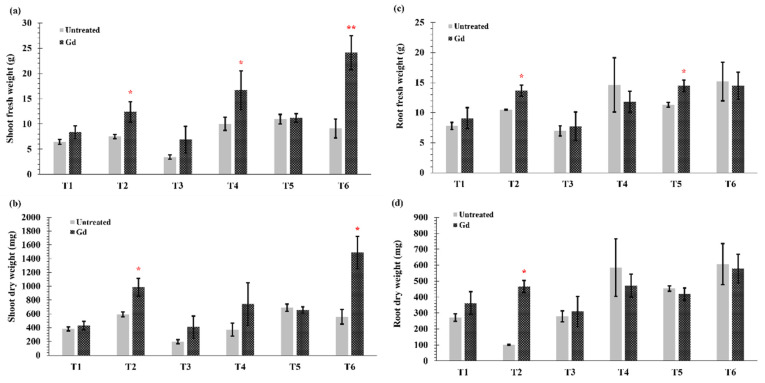
Evaluating the effects of severe water stress (35%-WHC; T1), moderate water stress (50%-WHC; T2), severe nitrogen stress (0 mg N pot^−1^; T3), moderate nitrogen stress (150 mg N pot^−1^; T4), combination of severe water stress and moderate nitrogen stress (35% WHC and 0 mg N pot^−1^; T5), and combination of moderate water stress and moderate nitrogen stress (50% WHC and 150 mg N pot^−1^; T6) on (**a**) Shoot fresh weight, (**b**) shoot dry weight, (**c**) root fresh weight, and (**d**) root dry weight of maize plants inoculated with *Gluconacetobacter diazotrophicus* Pal5 (Gd) in comparison with untreated plants. Bars represented means of three (*n* = 3) replicates with standard errors (SEs). * and ** above bars indicate significance at *p* < 0.05 and *p* < 0.01 and bars without any * are non-significant (*p* > 0.05).

**Figure 2 microorganisms-09-00870-f002:**
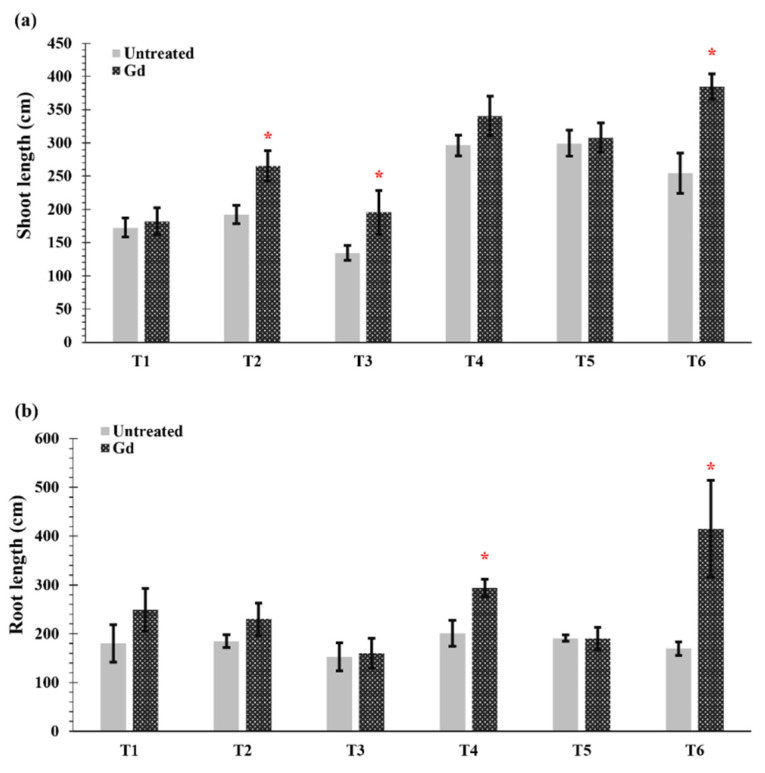
Evaluating the effects of severe water stress (35%-WHC; T1), moderate water stress (50%-WHC; T2), severe nitrogen stress (0 mg N pot^−1^; T3), moderate nitrogen stress (150 mg N pot^−1^; T4), combination of severe water stress and moderate nitrogen stress (35% WHC and 0 mg N pot^−1^; T5), and combination of moderate water stress and moderate nitrogen stress (50% WHC and 150 mg N pot^−1^; T6) on (**a**) shoot length and (**b**) root length of maize plants inoculated with *Gluconacetobacter diazotrophicus* Pal5 (Gd) in comparison with untreated plants. Bars represented means of three (*n* = 3) replicates with standard errors (SEs). * above bars indicate significance at *p* < 0.05 and bars without any * are non-significant (*p* > 0.05).

**Figure 3 microorganisms-09-00870-f003:**
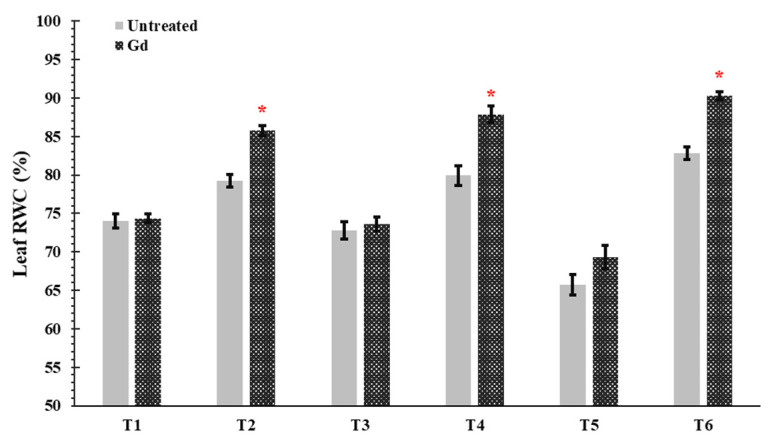
Evaluating the effects of severe water stress (35%-WHC; T1), moderate water stress (50%-WHC; T2), severe nitrogen stress (0 mg N pot^−1^; T3), moderate nitrogen stress (150 mg N pot^−1^; T4), combination of severe water stress and moderate nitrogen stress (35% WHC and 0 mg N pot^−1^; T5), and combination of moderate water stress and moderate nitrogen stress (50% WHC and 150 mg N pot^−1^; T6) on Leaf relative water contents (RWC) of maize plants inoculated with *Gluconacetobacter diazotrophicus* Pal5 (Gd) in comparison with untreated plants. Bars represented means of three (*n* = 3) replicates with standard errors (SEs). * above bars indicate significance at *p* < 0.05 and bars without any * are non-significant (*p* > 0.05).

**Figure 4 microorganisms-09-00870-f004:**
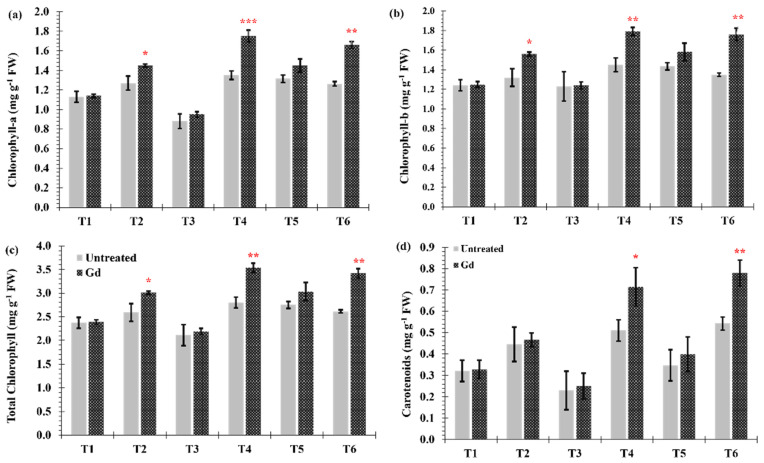
Evaluating the effects of severe water stress (35%-WHC; T1), moderate water stress (50%-WHC; T2), severe nitrogen stress (0 mg N pot^−1^; T3), moderate nitrogen stress (150 mg N pot^−1^; T4), combination of severe water stress and moderate nitrogen stress (35% WHC and 0 mg N pot^−1^; T5), and combination of moderate water stress and moderate nitrogen stress (50% WHC and 150 mg N pot^−1^; T6) on (**a**) Chlorophyll-a and (**b**) Chlorophyll-b, (**c**) total Chlorophyll and (**d**) carotenoids of maize plants inoculated with *Gluconacetobacter diazotrophicus* Pal5 (Gd) in comparison with untreated plants. Chlorophyll is shown in milligram per gram of fresh weight (mg·g^−1^ FW) of plants. Bars represented means of three (*n* = 3) replicates with standard errors (SEs). *, ** and *** above bars indicate significance at *p* < 0.05, *p* < 0.01 and *p* < 0.001, respectively. Bars without any * are non-significant (*p* > 0.05).

**Figure 5 microorganisms-09-00870-f005:**
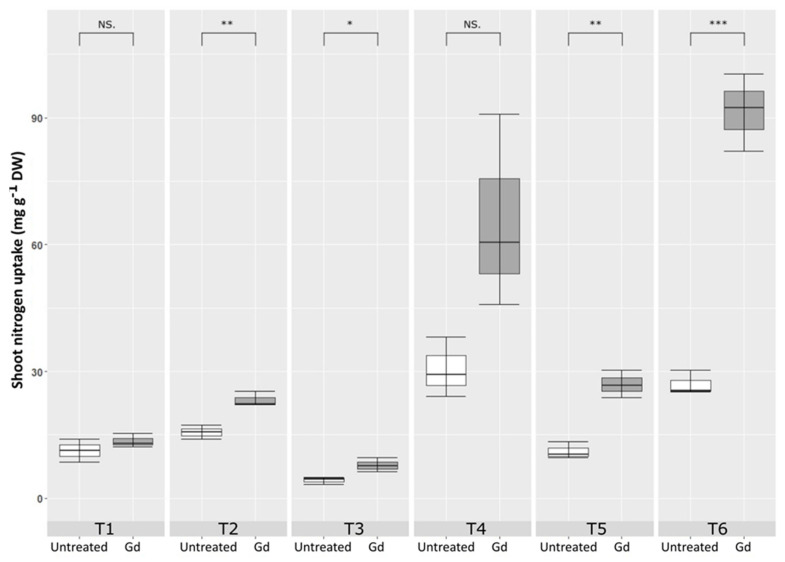
Evaluating the effects of severe water stress (35%-WHC; T1), moderate water stress (50%-WHC; T2), severe nitrogen stress (0 mg N pot^−1^; T3), moderate nitrogen stress (150 mg N pot^−1^; T4), combination of severe water stress and moderate nitrogen stress (35% WHC and 0 mg N pot^−1^; T5), and combination of moderate water stress and moderate nitrogen stress (50% WHC and 150 mg N pot^−1^; T6) on carotenoids of maize plants inoculated with *Gluconacetobacter diazotrophicus* Pal5 (Gd) in comparison with untreated plants. Shoot nitrogen uptake is given in milligram per gram of dry weight (mg·g^−1^·DW) of plants. Boxplots show the third quartile and first quartile (box edges), median (middle line) and range of the data (whiskers). Each boxplot represents the average of three samples. *, ** and *** above boxes indicate significance at *p* < 0.05, *p* < 0.01 and *p* < 0.001, respectively. Boxes without any * are non-significant (*p* > 0.05).

**Figure 6 microorganisms-09-00870-f006:**
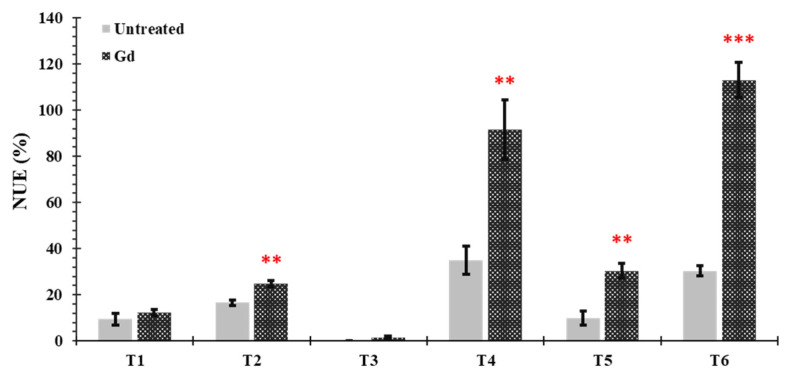
Evaluating the effects of severe water stress (35%-WHC; T1), moderate water stress (50%-WHC; T2), severe nitrogen stress (0 mg N pot^−1^; T3), moderate nitrogen stress (150 mg N pot^−1^; T4), combination of severe water stress and moderate nitrogen stress (35% WHC and 0 mg N pot^−1^; T5), and combination of moderate water stress and moderate nitrogen stress (50% WHC and 150 mg N pot^−1^; T6) on nitrogen use efficiency of maize plants inoculated with *Gluconacetobacter diazotrophicus* Pal5 (Gd) in comparison with untreated plants. Bars represented means of three (*n* = 3) replicates with standard errors (SEs). ** and *** above bars indicate significance at *p* < 0.01 and *p* < 0.001, respectively. Bars without any * are non-significant (*p* > 0.05).

**Figure 7 microorganisms-09-00870-f007:**
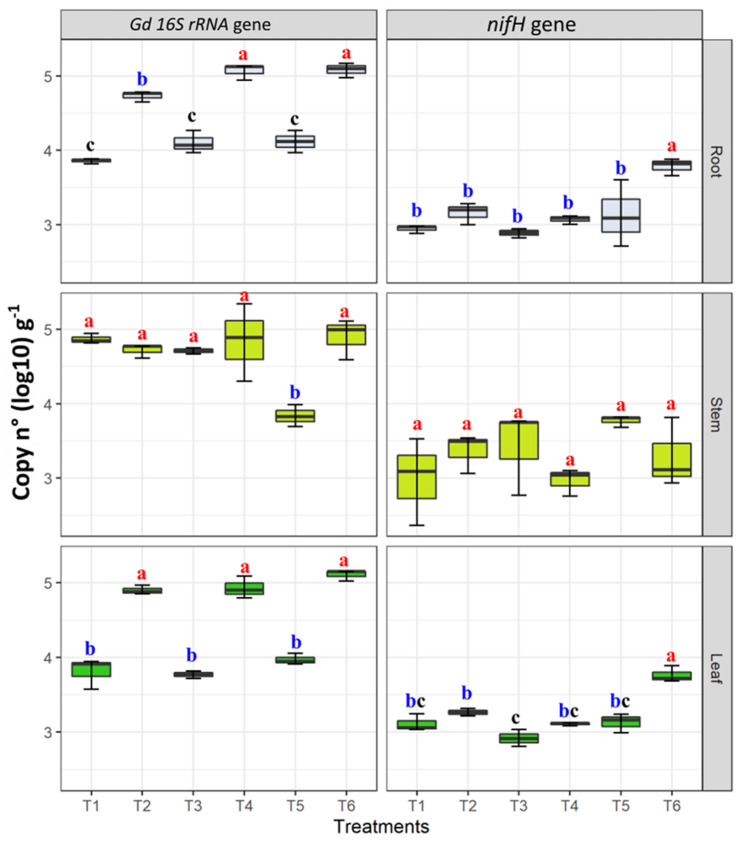
Quantification of *G. diazotrophicus* (Gd) *16S rRNA* and *nifH* gene copies in samples of maize plant tissues (leaves, stems, and roots) grown plants under water and nitrogen stress treatments, applied individually and in combination (T1–T6). Gene abundance was copy numbers log 10 per gram of fresh plant tissue (copy n° (Log 10)·g^−1^). Boxplots show the third quartile and first quartile (box edges), median (middle line) and range of the data (whiskers). Each boxplot represents the average of three samples. Boxplots having the same letters are significantly similar according to the Tukey HSD test at *p* < 0.05.

**Table 1 microorganisms-09-00870-t001:** Treatment plan.

Treatments	Description
T1	Soil moisture regime 35% of WHC with 100% nitrogen application
T2	Soil moisture regime 50% of WHC with 100% nitrogen application
T3	No nitrogen application with 100% WHC
T4	50% nitrogen application of recommended dose with 100% WHC
T5	Soil moisture regime 35% of WHC with 50% of nitrogen application
T6	Soil moisture regime 50% of WHC with 50% of nitrogen application

**Table 2 microorganisms-09-00870-t002:** The effects of water stress levels and N rates on plant water consumption (PWC), water use efficiency (WUE), and leaf rolling score of maize plants at the time of harvest. *, ** and *** indicate significance at *p* < 0.05, *p* < 0.01 and *p* < 0.001, respectively.

Treatments	PWC (mL)			WUE (mg/mL)			Leaf Rolling Score		
	Unt.	Gd	AOV	Unt.	Gd	AOV	Unt.	Gd	AOV
T1	719.7	747.1	*	0.35	0.39	ns	4.8	4.5	ns
T2	824.9	838.0	*	0.41	0.72	**	2.1	1.6	**
T3	1069.4	1100.6	ns	0.16	0.37	*	1.3	1.2	ns
T4	1120.2	1141.4	*	0.33	0.83	***	1	1	ns
T5	738.4	745.0	ns	0.41	0.49	ns	4.1	3.5	*
T6	809.0	840.5	**	0.69	1.99	***	2.2	1.5	**

## Data Availability

The data that support the findings of this study are available from the corresponding author upon reasonable request.

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
