# Peer review of "Gluconacetobacter diazotrophicus Pal5 Enhances Plant Robustness Status under the Combination of Moderate Drought and Low Nitrogen Stress in Zea mays L."

_microorganisms, 2021, doi:10.3390/microorganisms9040870_

Round 1
Reviewer 1 Report
The manuscript by Tufail et al. reports an interesting study on the role of Gluconacetobacter diazotrophicus Pal5 to mitigate the negative effects of drought and low N stress in maize. The results are reasonable and consistent with empirical knowledge. The data regarding the beneficial role of G. diazotrophicus Pal5 to alleviate the damaging effects of drought and low N supply can enrich our knowledge and valuable to be published. But right now, the manuscript is far away from accepting and has many things to be revised.
The most weak part of this manuscript is the materials and methods section, particularly the absence of positive well-watered control treatment as well lack of information about sampling and data collection. I do not get the idea behind the absence of positive control in the experimental treatments. It would have provided a better idea about how much increase in biomass and yield attributes is attained by microbial treatment of water stressed maize plants in comparison to control (well-watered plants).
Moreover, I cannot get detail and clear information which leaves were sampled for RWC and Chl measurements. The measurement of photosynthetic pigments content by using acetone is used only for rough determination, since it does not estimate chlorophyll b content correctly. The authors should provide much more details about green house conditions, i.e. light, temperature, humidity etc.
My major concern is the authors state that drought stress was initiated 12 days after sowing (line 120), and the seedlings were harvested 26 days after sowing (line 128). It is not possible that the plants would have experienced drought stress in such as short period i.e. only two weeks. The appearance of drought stress symptoms would have been key in this regard which take at least 3-4 weeks, particularly at moisture level of 50% WHC. This could be one of the reasons for non-significant results for this treatment. I would recommend to provide some data related to stress metabolites such as proline, glycine betaine, sugars, phenolics and antioxidants to get a better idea about the significant effect of microbial inoculation on maize performance under drought stress. In the absence of such data, the reliability of results is much lower.
Secondly, the authors used very small volume of soil (550 g) for pot, which could not be enough for 04 week old maize seedlings. In addition, I do not think that the application of even 50 mg N could be regarded as low N (50% N) as this amount of N would have been enough for 550 g soil since only plant is maintained in each pot, which is again questionable. Please provide the fertilisers application rates equivalent to recommended doses of N, P and K for maize.
All those information are critical to understand the results.
Author Response
Dear reviewer 1,
Speaking on behalf of all co-authors, I would like to acknowledge you for the constructive suggestions concerning our article entitled “Gluconacetobacter diazotrophicus Pal5 enhances plant robustness status under the combination of moderate drought and low nitrogen stress in Zea mays L.”. These suggestions on revision are valuable and helpful for improving our article. All the authors have worked on these suggestions and have tried to modify the manuscript for meeting the publication requirement of Microorganisms journal. Responses to your suggestions are listed below.
Response to Reviewer 1 Comments
The manuscript by Tufail et al. reports an interesting study on the role of Gluconacetobacter diazotrophicus Pal5 to mitigate the negative effects of drought and low N stress in maize. The results are reasonable and consistent with empirical knowledge. The data regarding the beneficial role of G. diazotrophicus Pal5 to alleviate the damaging effects of drought and low N supply can enrich our knowledge and valuable to be published. But right now, the manuscript is far away from accepting and has many things to be revised.
Point 1: The most weak part of this manuscript is the materials and methods section, particularly the absence of positive well-watered control treatment as well lack of information about sampling and data collection. I do not get the idea behind the absence of positive control in the experimental treatments. It would have provided a better idea about how much increase in biomass and yield attributes is attained by microbial treatment of water stressed maize plants in comparison to control (well-watered plants).
Response 1: We have tested the positive control using perlite as a growth medium in 300mm test tubes with two irrigation regimes including 50%-WHC and 100%-WHC. Results showed that G. diazotrophicus Pal5 inoculation significantly enhanced shoot and root weights and lengths and shoot nitrogen uptake of maize plants as compared to Untreated control at 50%-WHC (Fig.1 in this document), based on which we decided to minimize the well-watered control in this experiment. Additionally, our main objective was to evaluate the efficacy of G. diazotrophicus Pal5 only under individual and combined stress conditions. Since in nature the 100%-WHC does not occur predominantly and also due to the space and available resources required for this experiment.
Fig.1 (attached pdf) Evaluating the effects of 50%-WHC and 100%-WHC on shoot and root weights, lengths and endophytization with nitrogen contents in shoot of maize plants inoculated with Gluconacetobacter diazotrophicus Pal5 (Gd) in comparison with untreated plants. Boxplots show the third quartile and first quartile (box edges), median (middle line) and range of the data (whiskers). Each boxplot represents the average of five samples. * boxes indicate significance at p < 0.05 and boxes without any * are non-significant (p > 0.05).
Point 2: Moreover, I cannot get detail and clear information which leaves were sampled for RWC and Chl measurements. The measurement of photosynthetic pigments content by using acetone is used only for rough determination, since it does not estimate chlorophyll b content correctly. The authors should provide much more details about greenhouse conditions, i.e., light, temperature, humidity etc.
Response 2: Changes have been updated in the main text. We have added in the text precise information about the leaves which have been sampled to measure chlorophyll a and b. This information has been added in the manuscript (line= 144). Chlorophyll measurement was conducted following the comparison of Porra and Scheer (2019). These authors stated that chlorophyll measurement method by Mackinney G (1941) was antiquated/obsolete and it was updated by small changes in the equation and observation by Lichtenthaler and Wellburn (1983). Greenhouse conditions have been updated in the manuscript (line=115).
Point 3: My major concern is the authors state that drought stress was initiated 12 days after sowing (line 120), and the seedlings were harvested 26 days after sowing (line 128). It is not possible that the plants would have experienced drought stress in such as short period i.e. only two weeks. The appearance of drought stress symptoms would have been key in this regard which take at least 3-4 weeks, particularly at moisture level of 50% WHC. This could be one of the reasons for non-significant results for this treatment.
Response 3: As reported by Wang et al. (Wang et al., 2019), with different levels of nitrogen application, after two weeks of soil water regimes ended, they observed significant differences among several plant growth parameters such as leaf rolling, relative water contents, nitrogen use efficiency etc.
Point 4: I would recommend to provide some data related to stress metabolites such as proline, glycine betaine, sugars, phenolics and antioxidants to get a better idea about the significant effect of microbial inoculation on maize performance under drought stress. In the absence of such data, the reliability of results is much lower.
Response 4: Measuring these parameters is not possible at this stage as the aerial part and the roots of the plants have been used in their entirety to measure other parameters such as N content, chlorophyll, relative water contents, and molecular analysis. But your suggestion is very much appreciated, and we will certainly take it into account for our future work.
Point 5: Secondly, the authors used very small volume of soil (550 g) for pot, which could not be enough for 04 week old maize seedlings. In addition, I do not think that the application of even 50 mg N could be regarded as low N (50% N) as this amount of N would have been enough for 550 g soil since only plant is maintained in each pot, which is again questionable.
Response 5: As mentioned in the methodology, we used potting soil in small pots which is lower in weight and higher in volume than the field soil 500g is equal to 1.5 litre of potting soil as used by Egamberdieva et al (2013).
Point 6: Please provide the fertilisers application rates equivalent to recommended doses of N, P and K for maize.
Response 6: The nitrogen treatments consist of no-N (N-Free, 0 mg N pot-1) and 50% N (N-50, 150 mg N pot-1) and were applied in form of modified Hoagland solution, 7 days after sowing. Modified Hoagland solution was prepared using calcium nitrate, as a source of nitrogen fertilizer (McNickle et al., 2013). N-fertilization was calculated based on nitrogen, phosphorous and potassium (N-P-K:160-100-60 kg ha-1) recommended by Naveed et al. (2014) (line=129)
All those information are critical to understand the results.
References:
Egamberdieva D, Berg G, Lindström K, Räsänen LA. Alleviation of salt stress of symbiotic Galega officinalis L. (goat's rue) by co-inoculation of Rhizobium with root-colonizing Pseudomonas. Plant and Soil 2013; 369: 453-465.
Lichtenthaler HK, Wellburn AR. Determinations of total carotenoids and chlorophylls a and b of leaf extracts in different solvents. Portland Press Ltd., 1983.
Mackinney G. Abdorption of light by chlorophyll solutions. Journal of Biological Chemistry 1941; 140: 315-322.
McNickle GG, Deyholos MK, Cahill JF, Jr. Ecological implications of single and mixed nitrogen nutrition in Arabidopsis thaliana. BMC ecology 2013; 13: 28-28.
Naveed M, Mitter B, Reichenauer TG, Wieczorek K, Sessitsch A. Increased drought stress resilience of maize through endophytic colonization by Burkholderia phytofirmans PsJN and Enterobacter sp. FD17. Environmental and Experimental Botany 2014; 97: 30-39.
Porra RJ, Scheer H. Towards a more accurate future for chlorophyll a and b determinations: the inaccuracies of Daniel Arnon's assay. Photosynth Res 2019; 140: 215-219.
Wang Y, Zhang X, Chen J, Chen A, Wang L, Guo X, et al. Reducing basal nitrogen rate to improve maize seedling growth, water and nitrogen use efficiencies under drought stress by optimizing root morphology and distribution. Agricultural Water Management 2019; 212: 328-337.

Reviewer 2 Report
The manuscript describes the role of Gl. diazotrophicus Pal5 maize tolerance and growth under drought and nitrogen stresses. It is well-organized and each of the sections is well developed and detailed according to the Instructions to authors. Despite that, the abstract has to be more concise and shortened to 200 words. The literature is well synthesized in the Introduction but could be improved especially with studies about other nitrogen-fixing endophytes.
The questions set by the authors about the role of Gd population for the growth and development of maize’ earlier stages in a stressed environment were answered. The methodology is clearly explained and could be reproduced easily. The study possesses some novelty expressed in the development of primer for detection of the target bacterial population in maize. It is completely covering all stages of this kind of research, detecting and quantifying the Gd population in plant tissues.
The results are clear and easy to understand, while the discussion combines the results very well with the data from the literature in the field of the application of beneficial bacterial population.
Finally, in my opinion, the references have to be written all in the same way, with or without DOI number.
Author Response
Dear reviewer 2,
Speaking on behalf of all co-authors, I would like to acknowledge you for your constructive suggestions concerning our article entitled “Gluconacetobacter diazotrophicus Pal5 enhances plant robustness status under the combination of moderate drought and low nitrogen stress in Zea mays L.”. These suggestions on revision are valuable and helpful for improving our article. All the authors have worked on these suggestions and have tried to modify the manuscript for meeting the publication requirement of the Microorganisms journal. Responses to your suggestions are listed below.
Response to Reviewer 2 Comments
Point 1: The manuscript describes the role of Gl. diazotrophicus Pal5 maize tolerance and growth under drought and nitrogen stresses. It is well-organized and each of the sections is well developed and detailed according to the Instructions to authors. Despite that, the abstract has to be more concise and shortened to 200 words. The literature is well synthesized in the Introduction but could be improved especially with studies about other nitrogen-fixing endophytes.
Response 1: Thank you for taking the time to review our manuscript and providing your valuable feedback to improve the manuscript. Abstract has been modified accordingly. We added literature about other nitrogen fixing endophytes in the text (line=76).
Point 2: The questions set by the authors about the role of Gd population for the growth and development of maize’ earlier stages in a stressed environment were answered. The methodology is clearly explained and could be reproduced easily. The study possesses some novelty expressed in the development of primer for detection of the target bacterial population in maize. It is completely covering all stages of this kind of research, detecting and quantifying the Gd population in plant tissues.
Response 2: We appreciate your comments about the methodology. We really believe that a good materials and methods section can help our fellow scientists to better understand our work and to reproduce it. The fact that the journal microorganism does not have a maximum word limit means that this more descriptive section can be truly comprehensive and that all the necessary information is included.
Point 3: The results are clear and easy to understand, while the discussion combines the results very well with the data from the literature in the field of the application of beneficial bacterial population.
Response 3: We thank you for your appreciation.
Point 4: Finally, in my opinion, the references have to be written all in the same way, with or without DOI number
Response 4: Changes have been incorporated in the main text with all available DOI number.